# Self-Reported Outcomes of Endocrine Therapy with or Without Ovarian Suppression in Premenopausal Breast Cancer Patients: A Brazilian Quality-of-Life Prospective Cohort

**DOI:** 10.3390/cancers17193229

**Published:** 2025-10-04

**Authors:** Natália Nunes, Giselle Carvalho, Bernardo Ramos, Juliana Pecoraro, Lilian Lerner, Debora Azevedo, Thamirez Ferreira, Larissa Santiago de Moura, Carolina Galvão, Mariana Monteiro

**Affiliations:** 1Department os Clinical Oncology, Instituto Américas, Rio de Janeiro 22775-001, Brazil; giselle.medicinauerj@gmail.com (G.C.); bernardocmr@gmail.com (B.R.); julianapecoraro@institutoamericas.org (J.P.); lilianlerner@americasoncologia.com.b (L.L.); deboravictorino@institutoamericas.org (D.A.); thamirezvieira@institutoamericas.org (T.F.); larissamoura@institutoamericas.org (L.S.d.M.); carolgalvao@live.com (C.G.); 2Department os Clinical Oncology, Instituto Américas, São Paulo 22775-001, Brazil; marianamonteiro@institutoamericas.org; 3Department os Clinical Oncology, Hospital Samaritano, São Paulo 01232-010, Brazil

**Keywords:** quality of life, sexual dysfunction, survivorship, premenopausal women, endocrine therapy side effects, GnRH agonists, EORTC QLQ-BR23

## Abstract

Breast cancer is increasingly affecting young women, and many require long periods of treatment that may influence their quality of life. One important therapy for this group is endocrine treatment, sometimes combined with ovarian suppression. While these treatments improve cancer outcomes, they can also cause menopausal symptoms, affect body image, and interfere with sexual health. In this study, we followed premenopausal women with breast cancer in Brazil and collected quality-of-life data every three months for two years using standardized questionnaires. We found that women who received ovarian suppression reported more persistent difficulties, especially related to body image, compared with those treated with endocrine therapy alone. These findings emphasize the importance of considering not only cancer control but also quality of life when making treatment decisions and highlight the need for supportive care strategies tailored to young women.

## 1. Introduction

Breast cancer (BC) is the most common malignancy among women worldwide [1]. In Brazil and other Latin American countries, BC is proportionally more frequent in women under 40 years than in high-income countries (up to 17% vs. ~12% in North America) [2,3,4], partly explained by the younger age structure of Latin American populations. Recent studies, however, have reported a rising incidence among young women in several countries, including Brazil [4,5,6]. This younger population faces distinct clinical and psychosocial challenges compared with older women.

Among younger patients, estrogen receptor-positive (ER+) tumors predominate, accounting for the majority of cases [6]. Young women with BC frequently undergo chemotherapy, which can cause amenorrhea and ovarian failure in about 40% of women at age 40 and nearly 100% by age 50 [6,7,8]. In addition, while the use of adjuvant tamoxifen or aromatase inhibitors combined with ovarian function suppression (OFS) significantly reduces recurrence and mortality in early-stage disease [9,10,11], endocrine therapy (ET) is also associated with induced menopausal symptoms [12].

Menopausal symptoms triggered by ET and OFS are often more severe than those of natural menopause, including hot flashes, insomnia, vulvovaginal dryness, and reduced libido. These adverse effects can substantially impair quality of life (QoL) in young women and compromise treatment adherence [12,13,14,15]. Nevertheless, sexual health impairment remains a frequent but often overlooked consequence of cancer therapy, receiving limited attention from healthcare professionals during follow-up care. Sexuality is a multidimensional construct encompassing physical, psychological, interpersonal, social, and cultural domains [13,14,15,16,17]. As a result, survivors may experience substantial disruptions in sexual well-being, which should be addressed as part of comprehensive survivorship care to improve overall well-being and long-term outcomes [13,14,15,16,17].

Understanding the impact of ET on QoL across different cultural contexts is essential, as values and norms influence how patients perceive and cope with treatment side effects. However, data on patient-reported outcomes (PROs) from Latin America and other middle-income countries remain scarce. Generating context-specific evidence is crucial to informing care that addresses patients’ physical, psychosocial, and cultural needs.

This study aims to evaluate PROs in premenopausal patients with ER-positive BC treated with ET with or without OFS in the adjuvant setting in private healthcare facilities in Brazil. We hypothesized that adding OFS to ET would be associated with clinically meaningful worsening of sexual functioning and menopause-related symptoms compared with ET alone.

## 2. Materials and Methods

### 2.1. Study Design and Patient Population

This sub-analysis was derived from a multicenter, prospective, observational cohort conducted in Americas Oncologia, a private healthcare facility located in Rio de Janeiro and São Paulo, Brazil. In this analysis, we evaluated premenopausal women aged ≤50 years with stage I–III invasive estrogen receptor (ER)-positive BC. Patients with HER2-positive tumors were eligible if they were also ER-positive. Patients with stage IV disease, hormone receptor-negative tumors, or in situ carcinoma were excluded. All participants received adjuvant endocrine therapy (ET), with or without ovarian function suppression (OFS), between 1 January 2013 and 31 January 2023. Enrollment in the cohort occurred before initiation of systemic treatments.

PROs were assessed using the validated Portuguese version of the European Organization for Research and Treatment of Cancer Quality of Life Questionnaire—Breast Cancer module (EORTC QLQ-BR23) [18]. This 23-item questionnaire includes four functional and four symptom scales. Functional scales were scored so that higher values represent better functioning, whereas higher scores on symptom scales reflect greater symptom burden. For this analysis, the sexual functioning and sexual enjoyment scales were prespecified as primary outcomes. Selected symptom items (e.g., hot flashes, headache, fatigue) were also analyzed separately. Assessments were performed during in-person visits at baseline and at 3, 6, 9, 12, and 24 months. Questionnaires were administered by a trained research team that underwent calibration to ensure standardized data collection. Clinical, epidemiological, and disease-related variables were abstracted from medical records, while race/ethnicity was self-reported according to Brazilian census categories (Appendix Figure A1).

### 2.2. Statistical Analysis

Continuous variables were summarized as means with standard deviations (SDs), and categorical variables as absolute and relative frequencies. Associations between demographic/clinical variables and OFS use were examined using logistic regression.

Overall survival (OS) and disease-free survival (DFS) were estimated using Kaplan–Meier curves, which were compared with the log-rank test. Associations with OS and DFS were further evaluated with Cox proportional hazards models, reporting hazard ratios (HRs) with 95% confidence intervals (CIs). Both univariate and multivariate models were constructed, adjusting for age, stage, chemotherapy, radiotherapy, and type of surgery (mastectomy vs. breast-conserving).

EORTC QLQ-BR23 scores were summarized graphically over time and stratified by OFS status. Longitudinal changes in sexual functioning and sexual enjoyment were analyzed using linear mixed models, adjusted for cancer stage. A difference of >10 points was interpreted as clinically meaningful [19]. Results are expressed as means with 95% CIs, and between-group differences are presented for each time point.

For secondary analyses at the 24-month time point, categorical responses were dichotomized as “not at all/a little” versus “quite a bit/very much.” Comparisons between the OFS and ET-only groups were performed using Fisher’s exact test, given the relatively small sample size and sparse cells in some categories. *p*-values < 0.05 were considered statistically significant. All tests were two-sided. These analyses were exploratory and not adjusted for multiplicity.

Model assumptions were verified by evaluating the residuals of each regression model. Residual normality was assessed by visual inspection of histograms, which demonstrated approximately symmetric distributions centered around zero. All analyses were performed using R software, version 4.1.2 (R Foundation for Statistical Computing, Vienna, Austria). A two-sided *p*-value < 0.05 was considered statistically significant.

### 2.3. Ethical Considerations

The study protocol was reviewed and approved by the Research Ethics Committee (Comitê de Ética em Pesquisa—CEP of Hospital Pró-Cardíaco, Rio de Janeiro, Brazil) (approval date: 3 January 2012; reference no. 12747119.5.0000.5533). All participants provided oral and written informed consent. The study was conducted in accordance with Good Clinical Practice guidelines and the Declaration of Helsinki. The datasets generated and/or analyzed during the current study are available from the corresponding author upon reasonable request.

This prospective cohort was previously described in a 2018 publication addressing the clinical data and outcomes of the patients included until 2016, and also in a broader analysis of QoL outcomes in BC patients [19,20], all under the same institutional protocol and IRB approval (CEP Pró-Cardíaco, no. 12747119.5.0000.5533).

## 3. Results

A total of 363 female patients met the inclusion criteria. Most patients were white (50.1%), aged over 40 (66.0%), with a histological grade of 2 (53.0%), and treated with lumpectomy (53.0%) and (neo)adjuvant chemotherapy (77.0%). Only seven (1.9%) patients received adjuvant abemaciclib. The study population predominantly comprises patients with stage I and II BC, including 58.0% N0 patients, with only 17% stage III patients. HER2-positive patients comprised 17.0% of the cohort, and 56% had a Ki-67 index below 20% (Table 1).

For statistical analysis, two groups were identified: the ET-only group, comprising 290 patients (80.0%) who received ET alone, and the ET-OFS group, consisting of 73 patients (20.0%) who received ET combined with OFS. In multivariate logistic regression, younger age, advanced stage, and chemotherapy exposure were independently associated with OFS use. Compared with women under 35 years, those aged 35–40, 41–45, and 46–50 years had progressively lower odds of receiving OFS (OR 0.35, 95% CI 0.16–0.75, *p* = 0.007; OR 0.14, 95% CI 0.06–0.31, *p* < 0.001; and OR 0.07, 95% CI 0.03–0.16, *p* < 0.001, respectively). Patients with stage II (OR 2.38, 95% CI 1.27–4.63, *p* = 0.008) or stage III disease (OR 4.74, 95% CI 2.31–10.0, *p* < 0.001) were more likely to receive OFS. Chemotherapy exposure was also strongly associated with OFS use (OR 3.41, 95% CI 1.60–8.45, *p* = 0.003). In contrast, surgery type was not significantly associated, and patients on tamoxifen were less likely to receive OFS (OR 0.07, 95% CI 0.04–0.12, *p* < 0.001) (Table 2).

The responses to the EORTC QLQ-BR23 were obtained from 363 patients at baseline, 152 at 3 months, 272 at 6 months, 250 at 9 months, 252 at 12 months, and 171 at 24 months. Completion rates were similar between groups across time points (Appendix Table A1).

Analysis of sexual enjoyment and sexual functioning showed no statistically significant differences between ET-only and OFS-ET groups throughout follow-up. At baseline, mean scores were similar for enjoyment (72.2 vs. 76.1) and functioning (45.5 vs. 49.3). Both groups experienced declines during the first six months of treatment, with sharper reductions among OFS-ET patients. At 6 months, sexual enjoyment fell to 49.0 in the OFS-ET group compared with 58.3 in ET-only (between-group difference −4.67; 95% CI −9.94 to 0.61; *p* = 0.08), while sexual functioning declined to 30.7 vs. 34.9, respectively (difference −2.09; 95% CI −5.94 to 1.76; *p* = 0.29). These declines exceeded the threshold for clinical significance (>10 points) only in the OFS-ET group. Over time, ET-only patients largely recovered to baseline by 24 months (71.4 for enjoyment and 45.8 for functioning), whereas OFS-ET patients showed partial recovery but remained below baseline (68.6 and 44.7, respectively). Taken together, these findings indicate clinically meaningful but not statistically significant impairments in sexual health among women receiving OFS (Figure 1; Appendix Table A2, Table A3, Table A4 and Table A5).

At 24 months, no statistically significant differences were observed in sexual desire (51.5% vs. 42.0%; *p* = 0.33, Fisher’s exact test) or sexual enjoyment (26.0% vs. 13.5%; *p* = 0.20, Fisher’s exact test) between OFS and ET-only patients. Lack of sexual activity was more frequent in the OFS group (60.6% vs. 41.2%), with a borderline difference (*p* = 0.05, Fisher’s exact test). In contrast, body image outcomes were significantly more impaired among OFS patients, with higher proportions reporting feeling less attractive (38.2% vs. 19.9%; *p* = 0.04, Fisher’s exact test) and less feminine (26.5% vs. 11.7%; *p* = 0.05, Fisher’s exact test). These findings indicate that while sexual functioning domains showed numerical but nonsignificant differences, the impact on body image was more pronounced and statistically significant in patients undergoing ovarian suppression (Figure 2; Table 3; Appendix Table A6).

After a median follow-up of 37 months, 36 recurrences and 14 deaths were recorded. Median OS and DFS were not reached. Kaplan–Meier analysis showed no significant differences between ET-only and ET + OFS groups (log-rank *p* = 0.13; S8; Appendix Figure A2 and Figure A3; Appendix Table A7).

**Figure 2 cancers-17-03229-f002:**
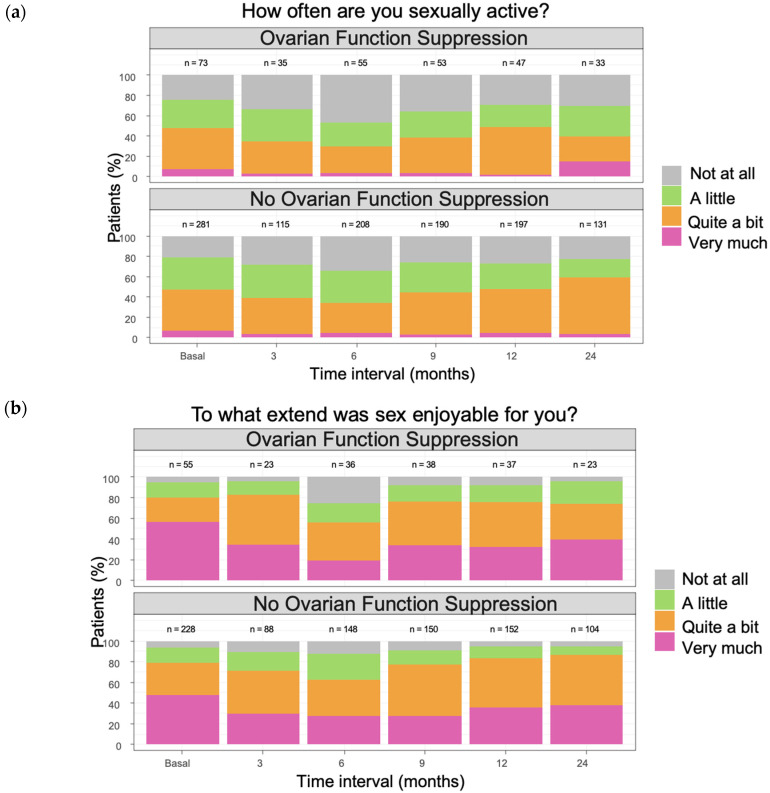
(**a**–**e**): Patient responses to the EORTC-QLQ-BR23: we present the distribution of responses to the selected items of the questionnaire, according to ovarian suppression, at each time interval.

**Table 3 cancers-17-03229-t003:** Sexuality and body image outcomes at 24 months. OFS = ovarian function suppression; ET = endocrine therapy.

Outcome	OFS (%)	ET-Only (%)	*p*-Value (Fisher)
Low sexual desire	17/33 (51.5%)	55/131 (42.0%)	0.33
Low sexual activity	20/33 (60.6%)	54/131 (41.2%)	0.05
Low sexual enjoyment	6/23 (26.1%)	14/104 (13.5%)	0.20
Body image—less attractive	13/34 (38.2%)	27/136 (19.9%)	0.04
Body image—less feminine	9/34 (26.5%)	16/137 (11.7%)	0.05

## 4. Discussion

Our study highlights the significant challenges faced by young BC patients undergoing endocrine therapy, particularly regarding sexual health, body image, and menopause-related symptoms. Patients receiving ovarian function suppression were younger and presented with more advanced disease, which underscores the complexity of treatment decisions: while OFS provides clinical benefits in high-risk populations [9,10,11], it also adds a substantial treatment-related burden. Although the differences were not statistically significant, patients in the ET-plus-OFS group experienced a persistent decline in sexual functioning and enjoyment, with scores not returning to baseline after 24 months. In contrast, scores among patients receiving ET alone eventually returned to baseline levels, suggesting that the addition of OFS may exacerbate long-term impairment in sexual health. OFS patients experienced clinically meaningful and persistent impairments, especially in body image.

Identifying gaps in QoL is critical for BC survivorship. Several studies [21,22] have shown a link between adherence to adjuvant ET and its impact on QoL, particularly in domains such as body image. For example, a recently published analysis [22] of Brazilian patients found that lower adherence was associated with poorer scores in the EORTC QLQ-BR23 body image domain. For instance, systematically monitoring and addressing these issues may not only improve patients’ QoL and sexual well-being but also positively influence treatment adherence and, ultimately, clinical outcomes.

Our results are consistent with the international literature. Sub-analyses of the SOFT and TEXT trials [23] confirmed worse vaginal dryness, reduced sexual interest, and difficulties in arousal among women receiving OFS, particularly in combination with aromatase inhibitors. The Mexican Joven & Fuerte cohort study [24] showed that 21.3% of young patients received ET plus OFS, and that sexual dysfunction rates increased from 33.6% at baseline to 52.9% at five years. The CANTO [14] cohort study, which followed nearly 8000 French women, found that 78.2% reported sexual concerns during follow-up. In this study, the use of ET was associated with poorer sexual functioning and sexual enjoyment. The use of OFS was not significantly related to these complaints. Still, only 2.3% of the patients had received it, which may have limited the ability to evaluate its effect on quality of life. Finally, a Norwegian study [25] demonstrated that BC survivors reported worse sexual functioning and enjoyment compared with non-cancer controls. The poorest outcomes were observed among premenopausal survivors and those treated with both chemotherapy and endocrine therapy. Together, these studies show that sexual dysfunction is prevalent, persistent, and probably insufficiently addressed in survivorship care. Our findings add to this evidence by providing data from Brazil, a middle-income country in Latin America.

This study has several limitations. The high proportion of patients treated with chemotherapy, even in the ET-only group, may have biased results, especially at 3- and 6-month time points, since chemotherapy itself induces amenorrhea, sexual dysfunction, and affects body image [7,8,12,13,19]. Similarly, the higher frequency of mastectomy in the OFS group may also have contributed to worse body image and perceptions of femininity [12,13,19]. Other differences in baseline characteristics between patients who received OFS and those treated with ET alone, such as younger age and more advanced disease, may be confounders that may compromise our ability to isolate the impact of OFS on QoL. These factors may also help explain why no significant differences in DFS and OS were observed between groups.

A noteworthy finding in our cohort is the relatively low proportion of patients who received OFS (only 20%), despite the high prevalence of chemotherapy exposure (77%). This is important, given that OFS is recommended for high-risk premenopausal patients, particularly those treated with chemotherapy [9,10,11]. Data from the SOFT and TEXT [9,10] trials, first reported in 2014, demonstrated that OFS combined with tamoxifen or aromatase inhibitors significantly reduced recurrence and mortality. Still, their uptake in practice in Brazil’s supplementary healthcare sector may have been gradual and heterogeneous. The limited use of OFS in our cohort may reflect physician preference, patient reluctance, or system-level barriers within Brazil’s supplementary healthcare sector. Still, it is crucial to take into account that the relatively small proportion of patients receiving OFS may have reduced the statistical power to detect differences between groups. Another potential limitation of our study is the long recruitment period (2013–2023), during which treatment strategies for premenopausal women with ER-positive BC evolved substantially worldwide. This temporal variability may also partly explain the relatively low proportion of patients receiving OFS in our cohort despite high rates of chemotherapy exposure. In addition, abemaciclib in the adjuvant setting was approved in Brazil in August 2021, near the end of our study period, and its incorporation was further limited by restricted coverage from private insurance providers. These factors account for the very low proportion of patients in our cohort who received adjuvant abemaciclib, despite it currently being considered the standard of care for high-risk ER-positive BC.

A critical observation in our study is the proportion of Black and Mixed-race women, who together accounted for about 32% of the cohort. At first glance, this may appear to be a substantial proportion of our cohort, especially considering that pivotal international trials such as SOFT and TEXT [9,10] did not report outcomes stratified by race, therefore limiting the generalizability of their findings to more diverse populations. However, considering the Brazilian context, where more than 55% of the population self-identifies as Black or Mixed race [26], this proportion of patients is lower than expected. This discrepancy raises concerns about representativeness and equity, as it underscores structural barriers that hinder access to supplementary healthcare for Black and Mixed-race patients. Structural inequities in Brazil’s private healthcare coverage have been well documented [27]. It is also important to note that Rio de Janeiro and São Paulo are located in the wealthiest region of Brazil, which further limits the generalizability of these data. In this context, similar studies evaluating PROs among patients treated in the public healthcare system (SUS) and across different regions of Brazil would be highly relevant to capture the impact of endocrine therapy and OFS in more socioeconomically and racially diverse populations.

As with any observational study, the possibility of other residual confounding cannot be excluded entirely. Taken together, these limitations should be considered when interpreting our results. Nevertheless, they do not diminish the relevance of our study, which provides real-world evidence from a middle-income country in Latin America and underscores the need for comprehensive survivorship care in premenopausal women with BC.

## 5. Conclusions

In conclusion, our findings emphasize the impact of ET, particularly when combined with OFS, on sexual health and QoL in young premenopausal BC patients in Brazil. Both groups experienced declines in sexual functioning and enjoyment, but those receiving OFS had a more persistent decline, with scores remaining below baseline after 24 months. Symptoms such as reduced sexual desire, diminished perceptions of femininity, and menopause-related complaints further illustrate the negative impact of these treatments on QoL despite their oncologic benefit.

These results highlight the urgent need for routine sexual health assessments, culturally sensitive interventions, and improved access to psychosocial and sexological support as integral components of survivorship care. Addressing these unmet needs could improve treatment adherence, enhance QoL, and ultimately lead to better outcomes for young breast cancer survivors.

## Figures and Tables

**Figure 1 cancers-17-03229-f001:**
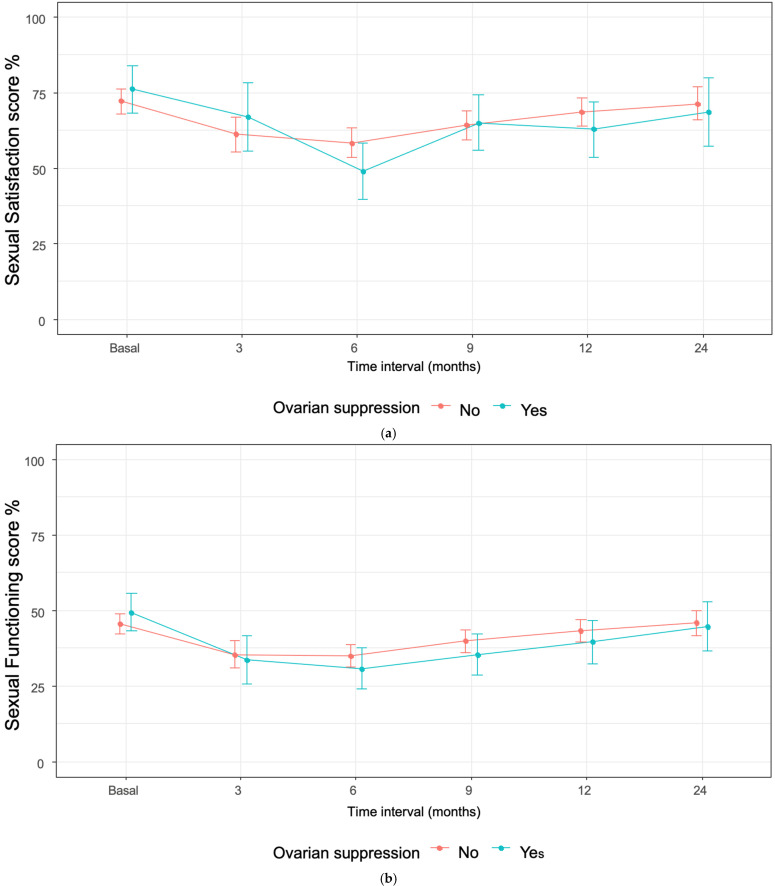
Sexual enjoyment (**a**) and sexual functioning (**b**) scales from the EORTC-QLQ-BR23 questionnaire. Patients under OFS showed a clinically significant decrease (greater than 10 points) in the 6-month timeline compared to baseline. At 24 months, patients in the ET-only group had returned to baseline, while patients in the OFS-ET group persisted with lower scores, although not considered clinically significant. No statistically significant differences were found between the groups.

**Table 1 cancers-17-03229-t001:** Baseline characteristics of patients.

Variables	Distribution, n/N (%)
Age at Diagnosis, n/N (%)	
<35 years	44/363 (12.0%)
Between 35 and 40 years	78/363 (21.0%)
Between 41 and 45 years	111/363 (31.0%)
Between 46 and 50 years	130/363 (36.0%)
Ethnicity, n/N (%)	
White	182/363 (50.1%)
Brown (Mixed Race)	62/363 (17.0%)
Black	25/363 (6.9%)
Asian	2/363 (0.6%)
Indigenous	1/363 (0.3%)
Missing	91/363 (25.1%)
Chemotherapy Treatment Received	279/363 (77.0%)
Adjuvant	187/363 (51.5%)
Neoadjuvant	96/363 (26.0%)
Ovarian Suppression	73/363 (20.0%)
Type of Surgery	
Lumpectomy	193/363 (53.0%)
Skin-Sparing Mastectomy	28/363 (7.7%)
Modified Radical Mastectomy	141/363 (39%)
Missing	1/363 (0.3)
Histological Grade	
G1	52/363 (14.0%)
G2	193/363 (53.0%)
G3	80/363 (22.0%)
Not Specified	36/363 (10.4%)
T Stage	
T1	185/363 (50.9%)
T2	119/363 (32.7%)
T3	46/363 (12.8%)
T4	10/363 (2.8%)
Tx	3/363 (0.8%)
N Stage	
N0	212/363 (58.0%)
N1	121/363 (33.0%)
N1mi	2/363 (0.6%)
N2a	16/363 (4.4%)
N2b	4/363 (1.1%)
N3a	3/363 (0.8%)
N3b	1/363 (0.3%)
N3c	3/363 (0.8%)
NX	1/363 (0.3%)
Stage	
I	44/363 (12.0%)
IA	102/363 (28.0%)
IB	2/363 (0.6%)
IIA	92/363 (25.0%)
IIB	60/363 (17.0%)
IIIA	47/363 (13.0%)
IIIB	9/363 (2.5%)
IIIC	7/363 (1.9%)
HER2 Status	
Positive	64/363 (17.6%)
Negative	296/363 (81.5%)
Missing	3/363 (0.8%)
Estrogen Receptor-Positive	354/363 (97.5%)
Progesterone Receptor-Positive	336/363 (92.5%)
Ki67 Index	
≤20%	191/363 (52.6%)
>20%	153/363 (42.1.0%)
Missing	19/363 (5.2%)

**Table 2 cancers-17-03229-t002:** Demographic and clinical characteristics by ovarian suppression. OR = Odds Ratio; CI = confidence interval.

	Total	Ovarian Suppression		Odds Ratio		
	N = 363	Yes, N = 73	No, N = 290	OR	95% CI	*p*-Value
Age, n/N (%)						
<35 years	44/363 (12.0%)	24/44 (55.0%)	20/44 (45.0%)	—	—	
35 to 40 years	78/363 (21.0%)	23/78 (29.0%)	55/78 (71.0%)	0.35	0.16, 0.75	0.007
41 to 45 years	111/363 (31.0%)	16/111 (14.0%)	95/111 (86.0%)	0.14	0.06, 0.31	<0.001
46 to 50 years	130/363 (36.0%)	10/130 (7.7%)	120/130 (92.0%)	0.07	0.03, 0.16	<0.001
Stage, n/N(%)						
I	148/363 (41.0%)	16/148 (11.0%)	132/148 (89.0%)	—	—	
II	152/363 (42.0%)	34/152 (22.0%)	118/152 (78.0%)	2.38	1.27, 4.63	0.008
III	63/363 (17.0%)	23/63 (37.0%)	40/63 (63.0%)	4.74	2.31, 10.0	<0.001
Type of Surgery, n/N (%)						
Lumpectomy	193/362 (53.0%)	34/193 (18.0%)	159/193 (82.0%)	—	—	—
Skin-Sparing Mastectomy	28/362 (7.7%)	4/28 (14%)	24/28 (86%)	—	—	—
Modified Radical Mastectomy	141/362 (39%)	34/141 (24%)	107/141 (76%)	1.91	0.68, 6.82	0.262
Chemotherapy, n/N (%)	279/363 (77.0%)	66/279 (24.0%)	213/279 (76.0%)	3.41	1.60, 8.45	0.003
Use of Abemaciclib, n/N (%)	7/363 (1.9%)	3/7 (43.0%)	4/7 (57.0%)	3.06	0.59, 14.2	0.149
Use of Tamoxifen, n/N (%)	285/363 (79.0%)	26/285 (9.1%)	259/285 (91.0%)	0.07	0.04, 0.12	<0.001

## Data Availability

The datasets generated and/or analyzed during the current study are available from the corresponding author upon reasonable request.

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
