# Peer review of "Self-Reported Outcomes of Endocrine Therapy with or Without Ovarian Suppression in Premenopausal Breast Cancer Patients: A Brazilian Quality-of-Life Prospective Cohort"

_cancers, 2025, doi:10.3390/cancers17193229_

Round 1

Reviewer 1 Report

Comments and Suggestions for Authors

The article examines the quality of life of breast cancer patients receiving hormone therapy, with or without ovarian suppression. The study explores how these treatments can increase toxicity and impair quality of life, with a specific focus on sexual health. The authors conclude that the reported symptoms significantly affect patients' sexual lives. While this finding is consistent with general medical literature, its relevance lies in being the first analysis of its kind conducted in a Brazilian population.

The text also points out the common barriers of lack of access to proper counseling and the embarrassment patients feel when addressing the issue. It emphasizes, however, that it's the responsibility of medical staff to provide appropriate guidance and refer patients to the correct specialists.

Key Points and Suggested Improvements

  • Long Study Period: The study, conducted between 2013 and 2023, spans a very long timeframe. The significant advancements in patient treatment and counseling in recent years could have influenced the results, limiting the validity of the conclusions over the entire decade.

  • Analytical Bias: The article mentions that patients with ovarian suppression feel less attractive. However, this finding could be biased, as a greater number of patients in this group underwent a mastectomy. This confounding variable is not adequately addressed in the conclusions, which weakens the analysis.

  • Weakness in Mortality Data: The conclusions regarding mortality are meaningless, as the sample size (300 patients) is too small to obtain statistically significant results. Furthermore, the fact that patients with ovarian suppression more frequently received chemotherapy introduces an additional confounding variable.

  • Treatment Limitations:

    • It's noteworthy that a high percentage of stage 3 patients did not receive hormonal blockade, despite being a high-risk population.

    • Only 1.7% of the patients received abemaciclib, likely because its approval was recent when the study concluded. Today, this treatment is the standard of care, and it would be crucial to analyze its impact on quality of life.

    • An adequate control group is missing, such as patients with HER2-enriched or triple-negative cancer who did not receive hormone therapy. This would allow for a more robust and comprehensive comparison.

Author Response

Comment 1: Long Study Period: The study, conducted between 2013 and 2023, spans a very long timeframe. The significant advancements in patient treatment and counseling in recent years could have influenced the results, limiting the validity of the conclusions over the entire decade.

Response 1: We thank the reviewer for this important comment. Given the evolving therapeutic landscape over the decade, we agree that the long recruitment period represents a limitation. We have explicitly acknowledged this point in the Discussion, noting that the gradual incorporation of ovarian suppression and the late approval of abemaciclib in Brazil may have influenced treatment patterns.

Comment 2: Analytical Bias: The article mentions that patients with ovarian suppression feel less attractive. However, this finding could be biased, as a greater number of patients in this group underwent a mastectomy. This confounding variable is not adequately addressed in the conclusions, which weakens the analysis.
Response 2: We agree with the reviewer’s observation. Indeed, mastectomy was more common in the OFS group and may have contributed to worse body image and perceived attractiveness. We have added this point as a potential confounder in the Discussion, clarifying that body image outcomes cannot be attributed solely to ovarian suppression.

Comment 3: Weakness in Mortality Data: The conclusions regarding mortality are meaningless, as the sample size (300 patients) is too small to obtain statistically significant results. Furthermore, the fact that patients with ovarian suppression more frequently received chemotherapy introduces an additional confounding variable.
Response 3: We acknowledge the reviewer’s concern. Our study was not powered to detect survival differences, as the primary endpoint focused on patient-reported outcomes and quality of life. Mortality results were reported descriptively and must be interpreted with caution. We have clarified this in the Discussion, stressing that survival analyses were limited by small numbers and confounding by chemotherapy exposure.

Comment 4: It's noteworthy that a high percentage of stage 3 patients did not receive hormonal blockade, despite being a high-risk population. Only 1.7% of the patients received abemaciclib, likely because its approval was recent when the study concluded. Today, this treatment is the standard of care, and it would be crucial to analyze its impact on quality of life.
Response 4: We appreciate this point. We have discussed the underutilization of ovarian suppression among high-risk stage III patients as a limitation, possibly reflecting physician and patient preferences or systemic barriers in Brazil’s supplementary healthcare system. Regarding abemaciclib, we clarified that its approval in Brazil only occurred at the end of our study period, which explains its very low uptake (1.9%). We agree that future studies are needed to assess its impact on quality of life.
Comment 5: An adequate control group is missing, such as patients with HER2-enriched or triple-negative cancer who did not receive hormone therapy. This would allow for a more robust and comprehensive comparison.
Response 5: We agree with the reviewer that including patients without endocrine therapy, such as those with HER2-enriched or triple-negative disease, would have provided a broader context and a more comprehensive comparison. However, the primary objective of this analysis was to specifically evaluate, within the ER-positive premenopausal population, the additional impact of ovarian function suppression compared to endocrine therapy alone. We fully acknowledge that assessing outcomes across groups with and without endocrine therapy could offer further insights into the true impact of ET, and we intend to explore this perspective in future analyses within the same cohort.

Reviewer 2 Report

Comments and Suggestions for Authors

The manuscript “Patient-reported outcomes in premenopausal non-metastatic breast cancer patients with or without ovarian suppression therapy: a subgroup analysis from a Brazilian prospective cohort” concentrated on reporting the most interesting results from assessments of patients' reported data (such as questionnaires) related to quality of life with special emphasis on sexual well-being. The results are mostly negative, however, data accumulation in this field is important for progress in real-life oncology, therefore, it would be logical to accept this manuscript for publication. Indeed, it may be used later for various meta-analyses. On the other hand, it has weaknesses stemming from the fact that data collected are limited to just one country.

MINOR:

“In Brazil, BC is the most frequently diagnosed cancer, excluding non-melanoma skin 39

Cancers”

Second most frequent?

“Furthermore, approximately 17% of BC patients in Brazil are younger than 40 42

Years”

is this different from any other countries?

“Although BC in younger patients tends to display more aggressive biological features, such as being estrogen receptor (ER) negative, the majority of tumors in these populations still express ER”

Strange sentence. It is widely recognized that these are different BC types.

Some of the phrases appear as if partially borrowed from other sources (just a few so I think that this is below plagiaris detection threshold), please corrects these:

  1. “encompassing physical, psychological, interpersonal, social, and cultural”.
  2. “only 11.4% of patients with poor sexual functioning received psychosocial counseling”

It is not clear how nationality/race is defined? Are there any criteria in Brazil? Is it self-reported?

“Although not statistically significant, our findings” – this sentence should be rewritten, findings cannot be “not significant”, rather, “not significant” should be referred to a difference between groups etc. And it would be nice to expose this more vividly.

Even from that abstract (“The OFS-ET group was younger (64.4% vs 25% <40 years) and more frequently treated with chemotherapy (90.4% vs 73.4%)” it is clear that the differences between groups may affect the results in such a manner that any conclusion is compromised. Perhaps, it would be good to include in the Discussion a statement about such limitations of this study.

Author Response

Comments 1: Comments about introduction: 

“In Brazil, BC is the most frequently diagnosed cancer, excluding non-melanoma skin 39

Cancers”

Second most frequent?

“Furthermore, approximately 17% of BC patients in Brazil are younger than 40 42

Years”

is this different from any other countries?

“Although BC in younger patients tends to display more aggressive biological features, such as being estrogen receptor (ER) negative, the majority of tumors in these populations still express ER”

Strange sentence. It is widely recognized that these are different BC types.

Some of the phrases appear as if partially borrowed from other sources (just a few so I think that this is below plagiaris detection threshold), please corrects these:

  1. “encompassing physical, psychological, interpersonal, social, and cultural”.
  2. “only 11.4% of patients with poor sexual functioning received psychosocial counseling”

Response 1: 

We thank the reviewers for these valuable observations. We agree that the Introduction, as originally written, lacked clarity and could be confusing in its structure. In the revised manuscript, we have thoroughly edited this section to ensure clearer contextualization and a more fluid narrative. Specifically, we:

  • Corrected the epidemiological statements to align with official Brazilian statistics and international comparisons.

  • Rephrased the sentence about ER expression in younger patients to avoid confusion.

  • Clarified the proportion of young patients in Brazil relative to other countries.

  • Reworded expressions that could appear borrowed, ensuring originality and fluency.

We hope these changes have improved both the accuracy and readability of the Introduction, making the background more coherent and informative for readers.

Comment 2: It is not clear how nationality/race is defined? Are there any criteria in Brazil? Is it self-reported?

Response 2: We thank the reviewer for raising this critical point. We have clarified in the Methods that, in Brazil, race/ethnicity is self-reported according to national census categories (white, Black, mixed-race, Asian, Indigenous). In addition, we expanded the Discussion to comment on the underrepresentation of Black and mixed-race patients in our cohort compared with the general Brazilian population, emphasizing the importance of reporting racial composition to highlight potential inequities in access to private oncology care.

Comment 3: Although not statistically significant, our findings” – this sentence should be rewritten, findings cannot be “not significant”, rather, “not significant” should be referred to a difference between groups etc. And it would be nice to expose this more vividly.
Response 3: We thank the reviewer for this stylistic observation. We revised the phrasing throughout the Results and Discussion to specify that the differences between groups were not statistically significant, rather than describing the findings themselves as “not significant.”

Comment 4: Even from that abstract (“The OFS-ET group was younger (64.4% vs 25% <40 years) and more frequently treated with chemotherapy (90.4% vs 73.4%)” it is clear that the differences between groups may affect the results in such a manner that any conclusion is compromised. Perhaps, it would be good to include in the Discussion a statement about such limitations of this study.
Response 4: We fully agree with the reviewer. In the revised Discussion, we highlighted that the ET+OFS group had younger patients, more advanced disease, and higher rates of chemotherapy and mastectomy, which may have influenced quality-of-life outcomes. We acknowledged these as important confounding factors that limit the strength of causal inferences in our study.

Reviewer 3 Report

Comments and Suggestions for Authors

Dear editor

The manuacript entitled"Patient-reported outcomes in premenopausal non-metastatic breast cancer patients with or without ovarian suppression therapy: a subgroup analysis from a Brazilian prospective cohort" evaluate patient-reported outcomes (PROs) in premenopausal patients with estrogen receptor-positive breast cancer undergoing ovarian function suppression compared to those who do not receive this treatment in the adjuvant setting.

This manuscript can be considered for publication after major revision and addressing following comments points by points:

1- Authors should exactly explain novelty of study .

2- Authors should design a schematic figure or graphical abstract to explain concisely the protocol.

3-In the section of "materials and methods", the commercial brands of materials or instruments should be mentioned.

4- Authours should improve the quality of Figures. The font size of legends, axis titles, axis values ,... should be increased

5-Result and discussion are poorly written. They should be improved.

6- Conclusion should be

improved.

Comments on the Quality of English Language

English should be polished.

Author Response

Comment 1: Authors should exactly explain the novelty of the study.
Response:
We thank the reviewer for this suggestion. We revised the Introduction and Discussion to highlight the novelty of our study better, emphasizing that it provides prospective patient-reported outcomes data from premenopausal ER-positive breast cancer patients treated in Brazil, a middle-income country, where real-world evidence is scarce.

comment 2: Authors should design a schematic figure or graphical abstract to explain concisely the protocol.
Response:
We appreciate this recommendation. We have prepared a schematic figure summarizing the study design, patient selection, and follow-up assessments, which has been added to the revised manuscript (Supplement figure 1).

comment 3: In the section of "materials and methods", the commercial brands of materials or instruments should be mentioned.
Response:
We thank the reviewer for this comment. Our study did not use laboratory reagents or commercial instruments. The only instrument employed was the validated Portuguese version of the EORTC QLQ-BR23 questionnaire, which has been appropriately referenced. We have clarified this point in the Methods section.

comment 4: Authors should improve the quality of Figures. The font size of legends, axis titles, axis values, … should be increased.
Response:
We agree with this suggestion and have improved the quality of all figures, increasing font size and resolution to enhance readability.

comment 5: Results and Discussion are poorly written. They should be improved.
Response:
We carefully revised both the Results and Discussion sections to improve clarity, structure, and flow, while ensuring that interpretations are balanced and supported by the data.

comment 6: Conclusion should be improved.
Response:
We have revised the Conclusion to better reflect the main findings of the study and their implications for survivorship care in young premenopausal breast cancer patients.

Reviewer 4 Report

Comments and Suggestions for Authors

The manuscript titled “Patient-reported outcomes in premenopausal non-metastatic breast cancer patients with or without ovarian suppression therapy: a subgroup analysis from a Brazilian prospective cohort”, authored by Natália Cristina Cardoso Nunes and colleagues, evaluates patient-reported outcomes in premenopausal patients with ER-positive breast cancer receiving ovarian function suppression compared to those who do not receive this treatment in the adjuvant setting.

The study addresses an important and often underexplored aspect of breast cancer management: the impact of endocrine therapy on sexual health and its influence on quality of life in premenopausal patients. The study is well-designed and properly conducted. The rationale is well articulated, highlighting the multidimensional significance of sexuality in patients’ lives and the influence of cultural values and norms on how women experience and manage the effects of endocrine therapy on their sexual health. The authors convincingly argue that exploring this issue is essential for designing comprehensive survivorship care that enhances overall well-being and long-term outcomes.

The manuscript is written in clear English, with appropriate terminology, and the statistical analyses are correctly applied in accordance with the hypotheses, variable types, and expected outcomes.

One limitation is that the study population was drawn from two private institutions in large Brazilian cities. Thus, the findings are more representative of urban populations and may not be generalizable to Brazil as a whole. Nevertheless, given the relatively large sample size and the extended study period (ten years), the research is valuable and makes a significant contribution to the field.

Lines 133–145 describe the characteristics of the two study groups (ET-only and ET-OFS) and their associations with clinical variables (tumor grade, cancer stage, chemotherapy, and type of surgical intervention). Women in the ET-OFS group had a higher frequency of grade 3 tumors, were more likely to present with advanced disease, more often received chemotherapy, and more frequently underwent mastectomy. Such differences are frequently reported in the literature as being associated with socioeconomic and ethnic disparities. Did the authors investigate whether such inequalities were present in their cohort? In this study, most women self-identified as white (50.1%), while those identifying as mixed-race, black, Asian, or Indigenous each accounted for less than 20% of the population. How does this distribution compare with the ethnic composition of the general populations of Rio de Janeiro and São Paulo? Given that participants were treated in private hospitals, their socioeconomic, educational, and cultural characteristics may not reflect those of the broader Brazilian population. While this limitation does not diminish the study's merit, a brief discussion of potential biases related to the study population would enable readers to contextualize the findings better.

In conclusion, this is a well-conducted study that makes an important contribution by drawing attention to sexual health issues in women with breast cancer treated with endocrine therapy. As a suggestion, including a brief discussion on the possible role of socioeconomic and ethnic disparities in shaping the observed differences between the two groups would provide a more comprehensive context for interpreting the results.

Author Response

comments : 

The manuscript titled “Patient-reported outcomes in premenopausal non-metastatic breast cancer patients with or without ovarian suppression therapy: a subgroup analysis from a Brazilian prospective cohort”, authored by Natália Cristina Cardoso Nunes and colleagues, evaluates patient-reported outcomes in premenopausal patients with ER-positive breast cancer receiving ovarian function suppression compared to those who do not receive this treatment in the adjuvant setting.

The study addresses an important and often underexplored aspect of breast cancer management: the impact of endocrine therapy on sexual health and its influence on quality of life in premenopausal patients. The study is well-designed and properly conducted. The rationale is well articulated, highlighting the multidimensional significance of sexuality in patients’ lives and the influence of cultural values and norms on how women experience and manage the effects of endocrine therapy on their sexual health. The authors convincingly argue that exploring this issue is essential for designing comprehensive survivorship care that enhances overall well-being and long-term outcomes.

The manuscript is written in clear English, with appropriate terminology, and the statistical analyses are correctly applied in accordance with the hypotheses, variable types, and expected outcomes.

One limitation is that the study population was drawn from two private institutions in large Brazilian cities. Thus, the findings are more representative of urban populations and may not be generalizable to Brazil as a whole. Nevertheless, given the relatively large sample size and the extended study period (ten years), the research is valuable and makes a significant contribution to the field.

Lines 133–145 describe the characteristics of the two study groups (ET-only and ET-OFS) and their associations with clinical variables (tumor grade, cancer stage, chemotherapy, and type of surgical intervention). Women in the ET-OFS group had a higher frequency of grade 3 tumors, were more likely to present with advanced disease, more often received chemotherapy, and more frequently underwent mastectomy. Such differences are frequently reported in the literature as being associated with socioeconomic and ethnic disparities. Did the authors investigate whether such inequalities were present in their cohort? In this study, most women self-identified as white (50.1%), while those identifying as mixed-race, black, Asian, or Indigenous each accounted for less than 20% of the population. How does this distribution compare with the ethnic composition of the general populations of Rio de Janeiro and São Paulo? Given that participants were treated in private hospitals, their socioeconomic, educational, and cultural characteristics may not reflect those of the broader Brazilian population. While this limitation does not diminish the study's merit, a brief discussion of potential biases related to the study population would enable readers to contextualize the findings better.

In conclusion, this is a well-conducted study that makes an important contribution by drawing attention to sexual health issues in women with breast cancer treated with endocrine therapy. As a suggestion, including a brief discussion on the possible role of socioeconomic and ethnic disparities in shaping the observed differences between the two groups would provide a more comprehensive context for interpreting the results.

Response:
We thank the reviewer for the positive assessment of our study. We fully agree that the representativeness of our cohort is limited, as participants were recruited from private hospitals in large Brazilian cities. This limitation has now been explicitly acknowledged in the Discussion. We also expanded the Discussion to highlight that the ET+OFS group differed from the ET-only group in terms of age, stage, chemotherapy use, and type of surgery, and that these baseline differences may reflect limits that influence treatment and quality-of-life outcomes. Regarding race/ethnicity, we clarified in the Methods that these data were self-reported according to Brazilian census categories. We further discussed that while 32% of participants identified as Black or mixed-race, this proportion remains lower than the national average (>55%), underscoring barriers to accessing private oncology care in Brazil.  We believe these additions provide a clearer context for interpreting our findings.

Reviewer 5 Report

Comments and Suggestions for Authors

The review of “Patient-reported outcomes in premenopausal non-metastatic breast cancer patients with or without ovarian suppression therapy: a subgroup analysis from a Brazilian prospective cohort” by Cardoso Nunes et al. studies the impact of endocrine therapy (ET), with or without ovarian function suppression (OFS), on the quality of life (QoL) of premenopausal women with estrogen receptor-positive (ER+) breast cancer in Brazil. It leverages a prospective, multicenter cohort and emphasizes patient-reported outcomes (PROs) regarding sexual functioning, sexual enjoyment, body image, and menopause-related symptoms.

The study addresses an important and underexplored issue: the effect of OFS on QoL in younger breast cancer patients. This is particularly relevant given the increasing incidence of breast cancer in younger women worldwide. The use of the EORTC-QLQ-BR23 questionnaire provides standardized and validated measures, allowing for robust evaluation of patient well-being. By focusing on a Brazilian cohort, the study contributes valuable data from a middle-income country context, where cultural and healthcare system factors may influence survivorship experiences.

Critical comments: The OFS-ET group represents only 20% of the cohort (73 patients), limiting statistical power and potentially underrepresenting the real impact of OFS in this population. Patients receiving OFS were younger and had more advanced disease, and more often received chemotherapy. These differences complicate the attribution of QoL outcomes solely to OFS. While clinically meaningful declines were observed in the OFS group, differences between groups were not statistically significant. The narrative sometimes emphasizes clinical trends over statistical evidence. Larger cohorts with better balance between ET-only and ET-OFS groups could have provided a more profound statistical background.

Overall, this article is a valuable contribution to the literature on young breast cancer survivors. It demonstrates that while OFS may improve oncologic outcomes in high-risk patients, it carries a substantial and sustained burden on QoL, particularly in sexual health. The study effectively calls for integrating routine sexual health assessments and culturally sensitive survivorship care into clinical practice.

Recommendations:

Please improve the quality of the presented figures, the resolution should be enhanced.

Please check the References section, because it contains information that was not written in English. The references should undergo revision.

Minor language revision should also be done to avoid inconsistencies in tense and style that slightly affect readability. 

Comments on the Quality of English Language

The manuscript is written in generally clear and comprehensible English. The structure is coherent, and scientific terminology is used appropriately. However, there are occasional grammatical errors, awkward phrasing, and minor inconsistencies in tense and style that slightly affect readability. These do not obscure the meaning but would benefit from light language editing by a native or professional scientific editor to enhance fluency and polish.

Author Response

Comment 1: The OFS-ET group represents only 20% of the cohort (73 patients), limiting statistical power and potentially underrepresenting the real impact of OFS in this population. Patients receiving OFS were younger and had more advanced disease, and more often received chemotherapy. These differences complicate the attribution of QoL outcomes solely to OFS.

Response: We acknowledge the imbalance in baseline characteristics and sample size between groups. This limitation has been emphasized in the revised Discussion section. We clarified that while our findings highlight clinically meaningful declines in QoL with OFS, the results should be interpreted with caution, given the potential confounding by age, stage, and chemotherapy exposure.

Comment 2: “While clinically meaningful declines were observed in the OFS group, differences between groups were not statistically significant. The narrative sometimes emphasizes clinical trends over statistical evidence.”

Response: Thank you for this observation. We revised the Results and Discussion sections to present clinical trends more cautiously and to distinguish between statistically significant findings and nonsignificant trends clearly. We also specified the statistical test used to enhance transparency.

Comment 3: “Please improve the quality of the presented figures, the resolution should be enhanced.”

Response: All figures have been replaced with high-resolution TIFF files, following the Reviewer’s requirements

Comment 4: Please check the References section, because it contains information that was not written in English. The references should undergo revision.”

Response: We carefully revised the References section and translated all non-English elements into English.

Comment 5: “Minor language revision should also be done to avoid inconsistencies in tense and style that slightly affect readability.”

Response: The entire manuscript has undergone an additional round of language editing to improve readability, ensure consistent use of tense, and harmonize style.

We believe that these revisions have improved the clarity and rigor of the manuscript. We thank the Reviewer for the constructive feedback, which greatly contributed to strengthening our work.

Reviewer 6 Report

Comments and Suggestions for Authors

1)    The title needs to be more clarified, i.e. to replace “patient-reported” with “self-reported”. Moreover “a subgroup analysis” is not necessary and can be removed. The main outcome is Quality of life (QoL), therefore it may appear mainly in the title rather that patient-reported outcome  

2)    The institution of authors should be reported extensively

3)    In the abstract how QoL was assessed should be mentioned. Results regarding logistic regressions in the abstract are lacking and need to be added

4)    The keywords should differs from those which appear in the title and the abstract, otherwise this becomes redundant and useless

5)    Following the aim in the Introduction section, a potential hypothesis should be added, in which authors mention an expected result or finding

6)    In the method section authors should mention how many articles have been published within this cohort by citing them, and if all used the same IRB number

7)    Authors should precise what is the exact name of the local Ethical Committee (i.e. website, email, phone number). Moreover the exact date of approval should be clearly mentioned.

8)    The authors should test for the distribution of their data, and report it. In case of abnormal distribution, Means and Standard deviations are not suitable

9)    The reporting of results is very poor, there is a lack of presentation regarding means as well as the logistic regression analysis

10) The discussion section is poorly presented. Is it possible that authors did not include any reference in this section; this reflects a poor quality of writing, that appears across the entire manuscript. Usually discussion section should include four main points:

  • The main finding of this study and an extensive comparison with previously published paper on the topic, in brazil as well as in other ethnic groups

  • The clinical implication of authors finding

  • The strength and limitations

  • The needed new directions for future research on the this specific topic  

11) The reference section is strongly unbalanced, out of 20 references in the entire manuscript, 17 are in the introduction section. 

Author Response

comment 1

The title needs to be more clarified, i.e. to replace “patient-reported” with “self-reported”. Moreover, “a subgroup analysis” is not necessary and can be removed. The main outcome is Quality of life (QoL), therefore it may appear mainly in the title rather than patient-reported outcome.

Response:
We thank the reviewer for this valuable suggestion. We revised the title to emphasize the main outcome (quality of life), replaced “patient-reported” with “self-reported,” and removed the phrase “a subgroup analysis.” We believe the new title is clearer and better reflects the study objectives.

comment 2

The institution of authors should be reported extensively.

Response:
We appreciate this point and have provided the full institutional affiliations of all authors in the revised manuscript. 

comment 3

In the abstract how QoL was assessed should be mentioned. Results regarding logistic regressions in the abstract are lacking and need to be added.

Response:
We agree with this comment and revised the Abstract to specify that QoL was assessed using the EORTC QLQ-BR23 questionnaire. We also included the key findings of the logistic regression analysis.

comment 4

The keywords should differ from those which appear in the title and the abstract, otherwise this becomes redundant and useless.

Response:
We thank the reviewer for this observation. We revised the keywords to avoid redundancy with the title and abstract.

comment 5

Following the aim in the Introduction section, a potential hypothesis should be added.

Response:
We agree with the reviewer and have added a hypothesis at the end of the Introduction, specifying our expectation that OFS combined with ET would result in more pronounced declines in QoL compared with ET alone.

comment 6

In the Methods section, authors should mention how many articles have been published within this cohort by citing them, and if all used the same IRB number.

Response:
We clarified in the Methods section that one previous publication has been derived from this cohort, under the same IRB approval, and cited it accordingly.

comment 7

Authors should specify the exact name of the local Ethical Committee (i.e., website, email, phone number). Moreover, the exact date of approval should be clearly mentioned.

Response:
We revised the Methods section to provide the full name of the Research Ethics Committee, the approval date, and reference number.

comment 8

The authors should test for the distribution of their data, and report it. In case of abnormal distribution, means and standard deviations are not suitable.

Response:
We appreciate this important methodological observation. As stated in the revised Methods, we verified model assumptions by visual inspection of histograms of residuals, which indicated approximate normality. 

Reviewer’s comment 9

The reporting of results is very poor, with a lack of presentation regarding means as well as the logistic regression analysis.

Response:
We revised the Results section to improve clarity and detail, including explicit presentation of means, standard deviations, and logistic regression findings.

Reviewer’s comment 10

The discussion section is poorly presented. It appears to lack references and does not follow the usual structure (main findings, comparison with previous studies, clinical implications, strengths and limitations, future directions).

Response:
We thank the reviewer for this valuable feedback. We have substantially revised the Discussion to address the main findings, compare them with national and international literature, discuss clinical implications, highlight strengths and limitations, and suggest directions for future research. References have been added to strengthen this section.

Reviewer’s comment 11

The reference section is strongly unbalanced, with most citations concentrated in the Introduction.

Response:
We revised and updated the References section to ensure a more balanced distribution across the manuscript, particularly in the Discussion.

Round 2

Reviewer 1 Report

Comments and Suggestions for Authors

After an in-depth review of the introduction, results, and discussion, I have achieved a significant improvement in the overall content and quality of the manuscript. I am confident that, with these enhancements, the article meets the criteria for publication.

Reviewer 3 Report

Comments and Suggestions for Authors

The manuscript is improved as well according to the comments and can be accepted in present form.

Reviewer 6 Report

Comments and Suggestions for Authors

.